# A Single Mutation Increases the Thermostability and Activity of *Aspergillus terreus* Amine Transaminase

**DOI:** 10.3390/molecules24071194

**Published:** 2019-03-27

**Authors:** Wan-Li Zhu, Sheng Hu, Chang-Jiang Lv, Wei-Rui Zhao, Hong-Peng Wang, Jia-Qi Mei, Le-He Mei, Jun Huang

**Affiliations:** 1College of Pharmaceutical Science, Zhejiang University of Technology, Hangzhou 310014, China; urnotzhuwanli@163.com; 2Department of Biological and Pharmaceutical Engineering, Ningbo Institute of Technology, Zhejiang University, Ningbo 315100, China; genegun@zju.edu.cn (S.H.); zwr166@sohu.com (W.-R.Z.); 3School of Biological and Chemical Engineering, Zhejiang University of Science and Technology, Hangzhou 310023, China; yangtzelv@zju.edu.cn (C.-J.L.); wanghongpeng@hotmail.com (H.-P.W.); 4Department of Chemical Engineering, University of Utah, Salt Lake City, UT 84102, USA; meijiaqi123@gmail.com

**Keywords:** amine transaminase, mutual information (MI), saturation mutagenesis, differential scanning fluorimetry (DSF), thermostability

## Abstract

Enhancing the thermostability of (*R*)-selective amine transaminases (AT-ATA) will expand its application in the asymmetric synthesis of chiral amines. In this study, mutual information and coevolution networks of ATAs were analyzed by the Mutual Information Server to Infer Coevolution (MISTIC). Subsequently, the amino acids most likely to influence the stability and function of the protein were investigated by alanine scanning and saturation mutagenesis. Four stabilized mutants (L118T, L118A, L118I, and L118V) were successfully obtained. The best mutant, L118T, exhibited an improved thermal stability with a 3.7-fold enhancement in its half-life (*t*_1/2_) at 40 °C and a 5.3 °C increase in *T*_50_^10^ compared to the values for the wild-type protein. By the differential scanning fluorimetry (DSF) analysis, the best mutant, L118T, showed a melting temperature (*T*_m_) of 46.4 °C, which corresponded to a 5.0 °C increase relative to the wild-type AT-ATA (41.4 °C). Furthermore, the most stable mutant L118T displayed the highest catalytic efficiency among the four stabilized mutants.

## 1. Introduction

Chiral amines are indispensable building blocks for numerous biologically active compounds and active pharmaceutical ingredients [1,2,3]. Transaminases are pyridoxal-5-phosphate (PLP)-dependent enzymes that can reversibly catalyze the transfer of an amino group from an amino donor to a prochiral ketone substrate [1,4]. In particular, amine transaminases (ATAs) have become the most prominent biocatalysts for the generation of optically pure chiral amines because of their high stereoselectivity and environmentally benign reaction conditions [2,5,6,7]. However, in industrial applications, biocatalysts, such as ATAs, are often required to enhance the rate of a reaction and the solubility of the reactant while simultaneously diminishing the risk of microbial contamination. The development of enzymes with a higher thermostability would expand the applicability of ATAs in industrial processes.

To enhance the thermal stability of an (*R*)-selective amine transaminase from *Aspergillus terreus* (AT-ATA), rational strategies, such as the combination of the B-factor profile and folding free energy calculations (ΔΔ*G*^fold^), the introduction of disulfide bridges, and consensus mutagenesis, were investigated [8,9,10]. Recently, the fact that a protein can be considered as a network of interacting residues has attracted great interest [11,12,13,14]. Specifically, nodes represent residues in a residue-residue coevolution network, and links are based on mutual information (MI), which is commonly based on Shannon’s entropy [15]. Residues are considered to be coevolving in the protein structure when a mutation at one position is compensated by a mutation at another position among homologous proteins [16,17]. An MI network can be developed to predict the location of different functional residues in protein families [17,18]. This strategy can not only reveal the relationship between the amino acid residues in a protein in terms of their structure and function but also make up for shortcomings in traditional directed evolution and rational design methods, effectively improving the efficiency of the molecular modification of enzymes. Hence, the analysis of MI networks is a promising method to guide the modification of proteins.

To minimize the effort and time invested in screening, various computational methods and the corresponding web-based tools have been used to identify mutation sites that are functionally or structurally important for proteins [19,20,21,22,23]. Compared to other publicly available web servers, the Mutual Information Server to Infer Coevolution (MISTIC) server provides integrated and concise insights into sequence and structure information in a visually rich manner. Particularly, the use of a circos representation of MI networks, the visualization of the cumulative MI (cMI), and proximity MI (pMI) concepts is original [23].

In this study, the MISTIC web server was used to predict the functional residues in the AT-ATA coevolution network. Subsequently, the effects of these functional residues on the enzymatic characteristics of AT-ATA were experimentally verified.

## 2. Results

### 2.1. Analysis of the Mutant AT-ATAs by Alanine Scanning

MISTIC provided an integrated view of AT-ATA in regards to the MI between residues (Appendix A), sequence conservation, cMI, and pMI. The cMI score and pMI score were both significant predictors of the functional residues of the protein. Residues with high cMI scores were rich in shared MI; residues with high pMI scores were mostly located in proximity to functionally important residues (distance < 5 Å) (Appendix A). High degrees of conservation at some residue positions indicated relatively lower rates of change during protein evolution. Further, an MI Circo was prepared (Appendix A), which is a circular representation of the mapped reference sequence using the information obtained from the MISTIC server [23]. Eight residues with important interactions were identified (Figure 1a) with MI, conservation, cMI, and pMI scores of 15, 1, 556, and 1984, respectively. We then performed an alanine scanning of these eight amino acid residues (H76, F115, E117, L118, N181, W184, T274, and T275). Among these substitutions, the variant L118A exhibited an enhanced thermostability in comparison to the wild-type enzyme with a nearly 3-fold increase in the half-life (*t*_1/2_) at 40 °C and a *T*_50_^10^ value of 42.2 °C (that of the wild-type enzyme was 38.5 °C). However, variants such as H76A, N181A, W184A, T274A, and T275A had decreased enzyme activities (Appendix A). As shown in Figure 1a, the residues mentioned above are mostly highly conserved; once these residues were replaced with alanine, the stable conformation was likely changed.

### 2.2. Saturation Mutagenesis of Residue L118

The saturation mutagenesis library of residue L118 was prepared with the goal of keeping the screening effort to a minimum and rapidly determining the hot spot for further engineering. Approximately 300 colonies were randomly selected from the saturation mutagenesis library. The results of the first round are shown in Appendix A. The enzymatic activities of the mutants before and after heating at 50 °C (for 10 min) are presented in Appendix A, respectively. Accordingly, the enzyme activity of each clone without heat treatment was assumed to be 100%, and the relative activities of the enzymes in the corresponding wells were calculated (Appendix A). The colonies with an apparently enhanced thermostability were isolated and verified by sequencing. Finally, four potential mutants (L118T, L118I, L118A, and L118V) were obtained.

### 2.3. Thermal Stability of AT-ATA and its Variants

As shown in Figure 2, the four purified variants (L118T, L118I, L118A, and L118V) showed a single band with the same molecular weight as the wild-type enzyme.

The inactivation of the three mutants (L118T, L118I, and L118V) from the saturation mutagenesis library was evaluated. As shown in Figure 3a, the *T*_50_^10^ values for the L118T, L118I, and L118V mutants increased to 43.8 °C, 41.6 °C, and 40.7 °C, which correspond to increases of 5.3 °C, 3.1 °C, and 2.2 °C over that of the wild-type enzyme (38.5 °C), respectively. In addition, the *t*_1/2_ values of the L118T, L118I, and L118V variants were 26 min, 18.4 min, and 15 min, respectively, compared to 6.9 min for the wild-type AT-ATA (Figure 3b). Notably, L118T displayed the largest improvement in thermal stability among the four variants. Additionally, the optimum catalytic temperatures of the L118I, L118V, and L118A variants were higher than that of the wild-type enzyme (Figure 3c).

The thermal unfolding of proteins was monitored by differential scanning fluorimetry (DSF) in the presence of SYPRO orange dye. When the protein unfolded, the hydrophobic parts were exposed to the dye. In DSF, the fluorescence intensity is plotted as a function of temperature (Appendix A). The transition midpoint value between the initial point and the maximum point can be identified as the melting temperature (*T*_m_). As shown in Table 1, L118A, L118I, L118V, and L118T showed higher *T*_m_ values (2.1, 1.5, 0.9, and 5.0 °C, respectively) than the wild-type enzyme.

### 2.4. Kinetic Constants

The kinetic constants of the AT-ATA and its variants were determined by monitoring the initial reaction rates at different concentrations of pyruvate and (*R*)-α-methylbenzylamine ((*R*)-α-MBA) as substrates, and the reaction was triggered by adding enzyme at the appropriate concentration. The Michaelis constants (*K*_m_^pyruvate^ for pyruvate and *K*_m_^α-MBA^ for (*R*)-α-MBA) and maximum velocity for the reaction (*V*_max_) were obtained from the Lineweaver–Burk plot, and the turnover number (*k*_cat_) and the catalytic efficiency (*k*_cat_/*K*_m_) were calculated (Table 2). The most obvious change in catalytic efficiency (*k*_cat_/*K*_m_^pyruvate^) was observed for the mutant L118T (2.22 to 4.85 s^−1^·mM^−1^), but this value decreased to 1.76 s^−1^·mM^−1^ for the mutant L118I. Noticeably, all four stabilized mutants showed higher values of *k*_cat_/*K*_m_^α-MBA^, and the mutant L118T showed the greatest increase in the *k*_cat_/*K*_m_^α-MBA^ value (from 2.82 to 9.98 s^−1^·mM^−1^). The results for *K*_m_^pyruvate^ revealed that the mutants had lower binding affinities for the substrate than the wild-type enzyme. In contrast, the results of *K*_m_^α-MBA^ indicated that all four stabilized mutants had even greater substrate-binding affinities than the wild-type enzyme. The most obvious change in the *K*_m_^α-MBA^ value was the decrease from 0.23 to 0.13 mM observed for the mutant L118I.

### 2.5. The Asymmetric Synthesis of α-MBA Catalyzed by AT-ATA and the Mutant L118T

The asymmetric synthesis of chiral amine α-MBA was performed using AT-ATA and the mutant L118T (Scheme 1). The standard samples (*R*)-α-MBA and (*S*)-α-MBA were derivatized with chiral Marfey’s reagent (*N*-α-(2,4-dinitrophenyl-5-fluorophenyl)-l-alanine amide, FDAA) to determine the configuration of the product by ultra-performance liquid chromatography coupled with electrospray ionization quadrupole time-of-flight tandem mass spectrometry (UPLC/ESI-QTOF-MS) on a Waters Acquity H-Class UPLC system coupled with a Xevo G2-XS mass spectrometer. As shown in Figure 4a, the extracted ion chromatogram (EIC) of the Marfey’s derivatives revealed the presence of FDAA with mass-to-charge (*m*/*z*) = 271, FDAA-2-butylamine with *m*/*z* = 324, and FDAA-(*R*)-α-MBA and FDAA-(*S*)-α-MBA with *m*/*z* = 372, but FDAA-(*R*)-α-MBA (1.61 min) and FDAA-(*S*)-α-MBA (1.81 min) had slightly difference retention times. In Figure 4b, the EIC of the Marfey’s derivatives revealed the presence of FDAA-l-alanine with *m*/*z* = 340 and FDAA-(*R*)-α-MBA and FDAA-(*S*)-α-MBA with *m*/*z* = 372; FDAA-(*R*)-α-MBA (3.85 min) and FDAA-(*S*)-α-MBA (4.24 min) showed different retention times. However, the (*S*)-α-MBA signal was not detected by the UPLC-MS analysis as a product of the transamination reaction. This result confirmed that the product generated by AT-ATA and the mutant L118T was (*R*)-α-MBA rather than (*S*)-α-MBA, and the substitution of Leu118 with Thr had no effect on the chiral selectivity of AT-ATA.

## 3. Discussion

To expand its industrial application for the production of chiral amines, most ATAs for biocatalysis need to be stabilized. In this study, we analyzed the MI for this protein family using the MISTIC server to identify the functional amino acid residues. The results of the alanine scanning indicated that L118A substitution resulted in an increase in the enzyme activity by approximately 2.5-fold compared to that of the wild-type protein (Appendix A). According to the crystal structure of AT-ATA, it follows a similar dual binding mode: The bulky substituent of the aromatic amine or ketone substrate and the α-carboxylate of the keto acid or amino acid are accommodated by the large binding pocket defined mainly by the hydrophobic residues His55, Tyr60, Phe115, Glu117, Leu182, and Trp184, and the methyl group was accommodated in the small binding pocket lined by Val62, Phe115, Thr274, Thr275, and Ala276 [24]. Surprisingly, residue 118 was adjacent to the large and small substrate binding pockets. The key residue Leu118 was subjected to saturation mutagenesis, and the mutant L118T was found to have a *T*_50_^10^ value that was 5.8 °C higher and a *t*_1/2_ value that was 3.8-fold higher at 40 °C compared to the corresponding values of the wild-type enzyme. To the best of our knowledge, the increase of 5.3 °C in the value of *T*_50_^10^ observed in this study is the largest increase that has ever been achieved through a single mutation (Appendix A). The present coevolution network of wild-type AT-ATA, determined in this study, makes it well-suited to low temperatures. When Leu118 was replaced with Thr118, the existing balance between structure and functional of the AT-ATA coevolution network was broken and the new network endowed it with a higher thermostability.

## 4. Materials and Methods

### 4.1. Molecular Biology Tools and Reagents

The AT-ATA cDNA sequence, including the *Nco*I and *Xho*I restriction sites, and all the primers used for mutagenesis were synthesized by General Biosystems (AnHui) Co., Ltd. (Chuzhou, China). The recombinant plasmid pET28a-AT-ATA was used as the template. PrimeSTAR^®^ Max DNA polymerase was acquired from Takara Biotechnology (Dalian, China). Lactate dehydrogenase (LDH), dimethyl sulfoxide (DMSO), l-alanine, acetophenone, 2-butylamine, imidazole, isopropyl-β-d-thiogalactopyranoside (IPTG), and Ni-IDA-Sefinose™ resin kits were purchased from Sangon Biotech Co., Ltd. (Shanghai, China). The SYPRO orange dye was purchased from Invitrogen (Carlsbad, CA, USA).

### 4.2. In Silico Analysis Procedure

The multiple sequence alignment (MSA) of the ATA family was uploaded from Pfam (Pfam accession PF01063). The reference sequence and structure were set as A3XII7_LEEBM/39–275 and PDB code 4CE5 [24], respectively. The complete alignment was performed by the MISTIC server (http://mistic.leloir.org.ar) [22]. After calculating the MI scores of the residues, a network was created in which the residues were nodes and a significant coevolutionary signal could be obtained through the links between those nodes (Appendix A). Finally, eight functionally important residues, shown in Figure 1a, were screened based on their conservation, MI, cMI, and pMI scores.

### 4.3. Alanine Scanning of the AT-ATA Coevolution Network

The eight residues of the AT-ATA coevolution network were analyzed by alanine scanning to identify their functional contributions. The sequences of all the mutagenic primers are displayed in Appendix A. The polymerase chain reaction (PCR) was carried out in an Applied Biosystems Veriti 96-well Thermal Cycler according to the method of Lin et al. [25]. The mixture consisted of PrimeSTAR Max Premix 2× (25 μL), pET28a(+)-AT-ATA plasmid (50 ng), forward and reverse primers (10 μM, 1 μL of each), and autoclaved water to a final volume of 50 μL. The PCR was cycled 30 times with the following program: 98 °C for 15 s (denaturation), 55 °C for 15 s (annealing), and 72 °C for 2 min (extension). The PCR products were digested with *Dpn* I (10 U; 37 °C, 3 h), purified, and then transformed into competent *E. coli* DH5α cells by the heat shock method. The transformed cells were incubated at 37 °C overnight on a Luria–Bertani (LB) agar plate containing 50 mg/L kanamycin. The recombinant plasmid DNA was isolated and verified by sequencing for further analysis.

### 4.4. Construction of the Saturation Mutagenesis Library

After alanine scanning mutagenesis, residue L118 was randomly replaced by saturation mutation, and the primers used to amplify the AT-ATA genes were 5′-GCATTTGTTGAANNNATAGT CACCCG-3′ (forward) and 5′-GCGGGTGACTATNNNTTCAACAAATGCATC-3′ (reverse). The PCR mixture for the saturation mutagenesis consisted of PrimeSTAR Max Premix 2× (25 μL), pET28a(+)-AT-ATA (1 μL), the forward and reverse primers (10 μM, 1 μL each), and sterile water to a final volume of 50 μL. The mixture was subjected to 1 cycle of 98 °C (1 min); 40 cycles of 98 °C (15 s), 55 °C (15 s), and 72 °C (1 min); and 1 cycle of 72 °C (7 min). The PCR products were treated with *Dpn*I and transformed into *E. coli* BL21 (DE3) cells by electroporation.

### 4.5. Library Screening

Nearly 300 colonies were randomly picked from the saturation mutagenesis library with sterile toothpicks and grown in a deep 96-well plate containing 1 mL LB medium with 50 mg/L kanamycin. When the optical density at 600 nm (OD_600_) reached 0.6–0.8, IPTG was added to each well at a final concentration of 1 mM. After the induction at 25 °C and the incubation for 20 h with shaking at 150 rpm, the 96-well plate was centrifuged at 4500 rpm for 10 min, the supernatant was discarded, and the cells were washed with buffer A (50 mM sodium phosphate buffer, pH 8.0) twice. Subsequently, the cells were resuspended in 200 μL of buffer a containing 5 mg/mL lysozyme and incubated at 37 °C for 30 min.

After centrifugation, 80 μL of the supernatant was removed from each well of the mother 96-well plate and transferred to the corresponding well of a daughter 96-well plate and heated at 50 °C for 10 min. After incubation, the plate was rapidly cooled in an ice bath and the activity of the enzyme was assayed at room temperature. The initial enzyme activity without heat treatment (control) was determined and used for a comparison. The mutants with higher residual activities than that of wild-type AT-ATA were further screened.

### 4.6. Protein Expression and Purification

One BL21(DE3) *E. coli* colony harboring wild-type AT-ATA or its variants in pET28a(+) was incubated in 2 mL of the LB medium containing kanamycin (50 mg/L) at 37 °C and 200 rpm overnight. The overnight culture was added to 200 mL of LB medium and then incubated at 200 rpm and 37 °C. When the OD_600_ reached approximately 0.6, the culture was incubated at 25 °C and 150 rpm in the presence of 1 mM IPTG. Subsequently, the cells were harvested by centrifugation at 6000 rpm for 10 min at 4 °C and washed twice with buffer A. The cells were resuspended in 30 mL of disruption buffer B (300 mM NaCl, 50 mM sodium phosphate buffer, 20 mM imidazole, and pH 8.0) and disrupted by ultrasonication for 15 min (sonication power, 300 W; duty time, 3 s; and interval time, 6 s). The intracellular protein was released, and the extract was centrifuged at 8000 rpm for 35 min at 4 °C; the supernatant was filtered through a 0.45-μm filter. The soluble recombinant proteins with an *N*-terminal His_6_-tag were purified by a Ni-affinity column. The purified proteins were eluted with elution buffer C (300 mM NaCl, 50 mM sodium phosphate, and pH 8.0) containing 250 mM imidazole. SDS-PAGE (12% separating and 5% stacking gels) was used to confirm the purity and molecular mass of the purified proteins. Furthermore, the concentration of the purified enzymes was estimated using a modified Bradford protein assay kit (Sangon Biotech Co., Ltd. Shanghai, China) [26].

### 4.7. Enzyme Assay and Kinetic Parameters

The enzyme activities of the wild-type enzyme and its mutants were investigated according to the method of Schätzle et al. [27]. Unless otherwise indicated, the transamination reaction of the AT-ATA and its mutants was measured by monitoring the production of acetophenone at 245 nm using a microplate reader (SpectraMax 190; Molecular Devices, Sunnyvale, CA, USA). Each measurement was repeated at least three times. The reaction mixture contained 2.5 mM *R*-MBA, 2.5 mM pyruvate, 0.25% (*w*/*v*) DMSO, 0.1 mM PLP in 180 μL of phosphate buffer A, and 20 μL of purified enzyme (0.1 mg/mL). Under these conditions, one unit of enzyme activity was defined as the amount of enzyme that generated 1 μmol of acetophenone per minute. The kinetic constants for (*R*)-α-MBA and pyruvate were determined by measuring the activities at different substrate concentrations when either pyruvate or *R*-MBA was held at 2.5 mM and the other substrate was present at 0, 0.125, 0.25, 0.5, 1.0, 1.5, 2.0, 2.5, or 3.0 mM. To calculate the kinetic constants, the obtained data were fitted to the Michaelis–Menten equation in Origin 8.0 (OriginLab Corp., Northampton, MA, USA).

### 4.8. The Thermal Stability of AT-ATA and its Mutants

#### 4.8.1. Kinetic Stability of AT-ATA and its Mutants

The *T*_50_^10^ value refers to the temperature at which the enzyme activity was reduced to half of the original activity after heat treatment for 10 min. The purified AT-ATA and its mutants were incubated for 10 min at temperatures from 25 to 55 °C in 5 °C increments and then cooled on ice for 10 min. The reaction was triggered by the addition of each enzyme, and the initial rate of the increase in absorbance at 245 nm was measured. The activity of the enzyme at 4 °C after 10 min was taken as 100%. To calculate the kinetic constants, the obtained data were fitted to the a four-parameter Boltzmann sigmoidal equation in Origin 8.0.

The *t*_1/2_ values of the AT-ATA and its mutants at 40 °C were determined by incubating the samples for various periods and then keeping them on ice for 10 min. The residual activity of each sample was detected at room temperature as described above. The activity of the corresponding enzyme without incubation at the same temperature was defined as 100%. The effects of temperature on the activity of the wild-type and mutant AT-ATA were determined by measuring the production of acetophenone at various temperatures for 3 min.

#### 4.8.2. Thermodynamic Stability of AT-ATA and its Mutants

Differential scanning fluorimetry (DSF) is an inexpensive and rapid method to identify the thermodynamic stability of purified enzymes. The wild-type enzyme and its mutants were analyzed using the method of Niesen et al. [28] with minor modifications.

The mixture consisted of 100 μg/mL pure enzyme, 1 × SYPRO Orange dye, buffer D (150 mM NaCl, 50 mM sodium phosphate, and pH 8.0), with a total volume of 50 μL. The samples with dH_2_O instead of pure enzyme were used as the negative control. Subsequently, the measurements were made on a StepOne Plus™ Real-Time PCR System (version 2.2.2, Applied Biosystems, Foster, CA, USA). Temperatures from 25 to 70 °C were scanned in increments of 0.7 °C, with each temperature maintained for 30 s. The fluorescence intensity was recorded with a 490 nm excitation wavelength and a 605 nm emission wavelength. The melting temperature (*T_m_*) of AT-ATA was calculated using the following equation:(1)$$y=UF+\frac{\left(NF-UF\right)}{1+e^{\frac{\left(T_{m}-x\right)}{a}}}$$
where *UF* and *NF* are the values of the minimum and maximum fluorescence intensities, respectively, and *a* denotes the slope of the curve within *T_m_*.

### 4.9. UPLC-MS Analysis of the Product of the Transamination Reaction

In the asymmetric synthesis of chiral amines catalyzed by AT-ATA and the mutant L118T, the selection of an appropriate amino donor and the elimination of the by-product inhibition were of great importance [29,30]. The sodium phosphate reaction buffer (50 mM, pH 8.0) containing 0.1 mM PLP was mixed with acetophenone (30 mM) and l-alanine or 2-butylamine (120 mM). The transamination reaction was triggered by the addition of purified enzyme coupled with LDH, and the mixture was shaken at 30 °C for 24 h.

The products of the transamination reactions catalyzed by AT-ATA and the mutant L118T were analyzed by UPLC-MS (Waters, Milford, MA, USA) according to the method of Xie et al. [10]. When 2-butylamine was used as an amino donor, the conditions for the UPLC analysis were as follows: an ACQUITY UPLC HSS T3 column (Waters, Milford, MA, USA) (2.1 × 100 mm, 1.8 μm) was used with a flow rate of 0.30 mL/min, and the mobile phase was a 35:65 (*v*/*v*) eluent A (0.1% formic acid in water (dH_2_O)) and eluent B (0.1% formic acid in acetonitrile). When l-alanine was used as the amino donor, the conditions for the UPLC analysis were as follows: an HSS T3 column was used, and the gradient elution conditions were 0–0.5 min (55:45), 3–4 min (35:65), and 4.5–5 min (55:45) at a flow rate of 0.30 mL/min. All samples were analyzed in negative ionization mode. The mass spectrometer conditions were as follows: cone voltage 40 V and ramp collision energy 20–55 V. The mass spectrum was acquired over the range of *m*/*z* 50–1200 Da.

## 5. Conclusions

A novel method involving an analysis of the MI to infer the functionally important residues was used to guide the molecular modification of AT-ATA. The *T*_50_^10^ and *t*_1/2_ values of the stabilized mutants at position 118 were greater than those of the wild-type enzyme by 2.2–5.3 °C and 8.1–19.1 min, respectively. As measured by DSF, the *T*_m_ value of the most stable mutant L118T reached 46.4 °C, which corresponded to an increase of 5 °C. Furthermore, the *k*_cat_ and *k*_cat_/*K*_m_ values of L118T were at least 2-fold higher than those of the wild-type enzyme. More importantly, the UPLC-MS analysis proved that the substitution of Leu118 with Thr had no effect on the chiral selectivity of AT-ATA. These results established a solid basis for future alterations to the design of ATA to further enhance its catalytic efficiency and thermostability, which is a prerequisite for applying this novel biocatalyst in the synthesis of chiral amines.

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
