# Peer review of "A Single Mutation Increases the Thermostability and Activity of Aspergillus terreus Amine Transaminase"

_molecules, 2019, doi:10.3390/molecules24071194_

Round 1

Reviewer 1 Report

See attached file

Author Response

20th March, 2019

Dear editor,

Thanks very much for your letter and the reviewers’ professional comments concerning our manuscript entitled "A single mutation increases the thermostability and activity of Aspergillus terreus amine transaminase" (Manuscript ID: molecules-465920). The reviewers’ comments were valuable and very helpful for clarifying and improving this manuscript. We have studied the comments carefully, and the manuscript has been revised thoroughly and carefully to correct language deficiencies. Revised portion are marked in red in the manuscript. In the following pages are our point-by-point responses to each comment of the reviewers. We hope that the corrections in the manuscript and our accompanying responses can meet with your approval and the manuscript suitable for publication in Molecules.

We are looking forward to hearing from you at your earliest convenience.

Yours sincerely,

Prof. Le-He Mei

Reviewer #1:

1.The authors should consider the possibility to extend the simulation time scale and the analysis on the trajectory structure at least up to 150 ns. If not, they should clearly mention this limitation and justify it. My major concern relates to the convergence of the 100ns MD simulation that indeed has not been thoroughly assessed. In fact, there is no evidence that a sort of equilibrium has been reached in the trajectory. The achievement of an adequate convergence in the MD simulations should be checked by calculating the root mean square inner product (RMSIP), i.e. for a 150 ns simulation between the two halves of the equilibrated trajectories (50-100 ns and 100-150 ns) considering the motion of Cα atoms along the first 10 eigenvectors [see Amadei, A., Ceruso, M. A. & Di Nola, A. On the convergence of the conformational coordinates basis set obtained by the essential dynamics analysis of proteins' molecular dynamics simulations. Proteins 36, 419-424 (1999)].

Response 1:  We thank the reviewer for raising this critical issue. As pointed out, the current molecular dynamics simulation results are indeed insufficient for this article. Considering the accuracy and completeness of this article, these uncertain contents has been removed in our revised manuscript, but this does not affect the novelty of the total paper. We would like to express our sincere appreciation for your careful reading and helpful comments.

2.The authors are kindly requested to specify which is the reference sequence that has been provided as input to the MISTIC server.

Response 2: The reference sequence is A3XII7_LEEBM/39-275.

3.What is the meaning of “intensive interaction residues”. Please specify.

Response 3: The “intensive interaction residues” means that “residues which are associated with multiple other residues in a coevolutionary network”.

4.In Figure S5, the error bars have been omitted.

Response 4: Thank the reviewer for pointing out the questions. The error bars have been added in the revised manuscript, as shown in Figure S5.

Figure S5. Thermal unfolding of wild-type AT-ATA and its mutants were monitored by DSF.

5. Overall, the written English is rather poor and the authors are kindly advice for a substantial editing carried out by a native English speaker.

Response 5: Thanks for your suggestions. We feel very sorry for our poor writings.  According to the reviewer’s comments, we have made the manuscript be edited by a native English-speaking editor. All changes were marked in red in the revised manuscript. And we hope that the revised manuscript could meet with your approval. The certificate of editing is as follows:

6. The following few sentences just represents typical examples:

Abstract, lines 28 and 29:

“Furthermore, molecular dynamic (MD) simulation demonstrated that the mutant L118T lowered the overall root mean square deviation (RMSD) and consequently increased protein rigidity.”

Response 6: Special thanks for the pertinent comments. In the revised manuscript, this inaccurate sentence of Furthermore, molecular dynamic (MD) simulation demonstrated that the mutant L118T lowered the overall root mean square deviation (RMSD) and consequently increased protein rigidity. has been removed.

7.Now, MD simulations DO NOT DEMONSTRATE anything, MD simulations may instead unveil a structural feature such as flexibility, etc.

“… lowered the overall RMSD”

RMSD of which parameter: the Ca position? Please specify.

Response 7: Thank the reviewer for pointing out the inaccurate statement. We have deleted the section “molecular dynamics simulation”.

8.Main text line 45

“…such as combination of B-factor coupling with folding free energy calculations…”

should read …such as combination of the B-factor profile and folding free energy calculations…

Response 8: We are grateful to the reviewer for correcting our inaccuracies. In the revised manuscript, this sentence has been revised to “To enhance the thermal stability of an (R)-selective amine transaminase from Aspergillus terreus (AT-ATA), rational strategies, such as combination of the B-factor profile and folding free energy calculations (ΔΔGfold), the introduction of disulfide bridges, and consensus mutagenesis, were investigated”.

9.Main text lines 165-166 It is hard to believe that MD simulations make it possible to obtain insights into the molecular mechanisms. Moreover, the associated Reference 24 seems to be not appropriate.

Response 9: We thank the reviewer for the valuable comments. In the revised manuscript, the inaccurate sentence “MD simulation makes it possible to investigate the conformation of proteins and obtain insights into the molecular mechanisms [24]” has been removed.

Reviewer 2 Report

In the second revision, the dubious claim of hydrogen bonding has been removed and the phrasing of MISTIC descriptions has been changed as requested. The crystal structure of ATA is still not referenced until the very end. It should be called out when it first appears in the figure legend of Fig. 1. Also, the use of reference 29 on line 329 appears to be incorrect. Other than that, I am satisfied that the authors have adequately responded to all issues raised in review.

Author Response

(The authors gave the same response as above.)

Round 2

Reviewer 1 Report

The authors have addressed the reviewer's comments and the manuscript is acceptable for publication in this journal.

This manuscript is a resubmission of an earlier submission. The following is a list of the peer review reports and author responses from that submission.

Round 1

Reviewer 1 Report

Title: A single mutation increases the thermostability and activity of Aspergillus terreus amine transaminase

This manuscript presents mutants of ATA that increase thermostability and catalytic efficiency. This is a significant finding with respect to using these enzymes industrially. The approach and experimentation are sound for the most part. However, there are several issues with the presentation of the data that require revision. In particular, the structural modeling presented in Fig. 7 is not well defined and the conclusions from this are highly speculative (see point 3).

Major revisions:

1.     Figure 3C would be better presented as actual initial rates at various temperatures rather than a relative measure. This would allow a direct comparison of rates between mutants and WT under the same conditions rather than just a profile of relative activity vs. temperature within each sample.

2.     The reaction and data presented in section 2.5 is not well described. Abbreviations are not defined, and it would be very helpful to have a scheme of the reaction. Also, why were whole cell extracts used rather than purified proteins? If whole cell extracts are used, the “Blank” should consist of whole cell extracts from E. coli cells not expressing the ATA to discount any native E. coli enzyme activity.

3.     The second sentence of the Discussion states “As detailed information of AT-ATA is not available...”. However, a crystal structure for this enzyme does exist, and it was used in the MD simulations. This should be stated with a reference at the beginning of section 2.6. Also in the Discussion and in Fig. 7, the authors identify H-bonds in the WT and L118T ATA. However, here again it is not clear that the existing WT crystal structure is used. Further, the only information I can find on the mutant model is that it was “created by PyMol”. That makes it seem that the authors simply built a Thr in place of the native Leu. That would suggest that this is not an energy minimized model and that the orientation of the Thr side chain is purely arbitrary, making the claim that an extra H-bond is formed extremely speculative. Further, it must be noted that L118A, L118I and L118V exhibit enhanced stability that obviously cannot be attributed to H-bonding. In general, structural elements of the discussion are very speculative and not well supported. The crystal structure of AT-ATA is finally mentioned in the last paragraph but is still not cited.

4.     In the Intro or beginning of Results, it would be nice to have a brief description of what data the MISTIC provides and the meaning of the various scores. In Fig. 1, what is meant by “lesser” or “higher” degree of conservation?

Minor revisions:

1.     P. 1, first paragraph: Delete the phrase “required by them” at the end of the 3rd sentence. The following sentence is vague and possibly unnecessary.

2.     P. 2, last sentence: Add the word “likely” to read “...the stable conformation was likely changed” since you cannot say this for a fact.

3.     P. 5: The first sentence of section 2.5 is a fragment.

4.     Numerous examples exist of abbreviations being used without a definition. Please define abbreviations on their first appearance.  

Author Response

Reviewer #1: 

Title: A single mutation increases the thermostability and activity of Aspergillus terreus amine transaminase

This manuscript presents mutants of ATA that increase thermostability and catalytic efficiency. This is a significant finding with respect to using these enzymes industrially. The approach and experimentation are sound for the most part. However, there are several issues with the presentation of the data that require revision. In particular, the structural modeling presented in Fig. 7 is not well defined and the conclusions from this are highly speculative (see point 3).

Major revisions:

1. Figure 3C would be better presented as actual initial rates at various temperatures rather than a relative measure. This would allow a direct comparison of rates between mutants and WT under the same conditions rather than just a profile of relative activity vs. temperature within each sample.

Response 1: Thank the reviewer for the suggestion. A profile of relative activity vs. temperature has been used in some literatures (Zhou et al. 2018; Huang et al., 2017; Xie et al., 2018). In the future work, actual initial rates at various temperatures will be used. kcat represents the initial rate,and kcat/Km stands for the catalytic efficiency of enzyme. The comparison of kcat and kcat/Km between mutants and WT under the same conditions have been performed in Table 1.

Huang J, Xie DF, Feng Y. Engineering thermostable (R)-selective amine transaminase from Aspergillus terreus through in silico design employing B-factor and folding free energy calculations. Biochemical and Biophysical Research Communications, 2017, 483(1): 397-402. 

Zhou Z, Li M, Xu JH, et al. A single mutation increases the activity and stability of Pectobacterium carotovorum nitrile reductase. ChemBioChem, 2018, 19(5): 521-526.

Xie DF, Fang F, Mei JQ, et al. Improving thermostability of (R)-selective amine transaminase from Aspergillus terreus through introduction of disulfide bonds. Biotechnology and Applied Biochemistry, 2018, 65(2): 255-262.

2.The reaction and data presented in section 2.5 is not well described. Abbreviations are not defined, and it would be very helpful to have a scheme of the reaction. Also, why were whole cell extracts used rather than purified proteins? If whole cell extracts are used, the “Blank” should consist of whole cell extracts from E. coli cells not expressing the ATA to discount any native E. coli enzyme activity.

Response 2: Thank the reviewer for pointing out our inaccurate statement. In the revised manuscript, the section 2.5 are marked in red. The new experiments catalyzed by purified AT-ATA have been performed, and the data of transamination reactions added (Figure 5). A scheme of the reaction has also been provided as follows:

Scheme 1. The asymmetric synthesis of chiral amines using AT-ATA  

2.The second sentence of the Discussion states “As detailed information of AT-ATA is not available...”. However, a crystal structure for this enzyme does exist, and it was used in the MD simulations. This should be stated with a reference at the beginning of section 2.6. Also in the Discussion and in Fig. 7, the authors identify H-bonds in the WT and L118T ATA. However, here again it is not clear that the existing WT crystal structure is used. Further, the only information I can find on the mutant model is that it was “created by PyMol”. That makes it seem that the authors simply built a Thr in place of the native Leu. That would suggest that this is not an energy minimized model and that the orientation of the Thr side chain is purely arbitrary, making the claim that an extra H-bond is formed extremely speculative. Further, it must be noted that L118A, L118I and L118V exhibit enhanced stability that obviously cannot be attributed to H-bonding. In general, structural elements of the discussion are very speculative and not well supported. The crystal structure of AT-ATA is finally mentioned in the last paragraph but is still not cited.

Response 3:  Thank the reviewer for pointing out our inaccurate statement in the manuscript. In the revised manuscript, this sentence “the crystal structure of AT-ATA (PDB ID: 4CE5) from the Protein Data Bank was used as a template [29], and the three-dimensional (3D) structures of thermostable AT-ATA mutants were homologically modeled and energy minimized using the software FoldX (version 3.0 beta5.1).” was added in the section 4.10. We have updated the data in Figure 7(a) and Figure 7(b), and the revised section in the discussion are marked in red.

4.  In the Intro or beginning of Results, it would be nice to have a brief description of what data the MISTIC provides and the meaning of the various scores. In Fig. 1, what is meant by “lesser” or “higher” degree of conservation?

Response 4:  Thank the reviewer for good suggestion. In the revised manuscript, the Results of MISTIC has been complemented as follows:

  MISTIC provided an integrated view of AT-ATA in terms of the mutual information between residues (Figure S1A), sequence conservation and cumulative Mutual Information (cMI) that measures the degree of shared mutual information of a given residue and the proximity Mutual Information (pMI), which tells about the networks of mutual information in the proximity of a residue (distance < 5 Å) (Figure S1B). The high conservation degree at some residue positions stands for a relatively lower rate of change during protein evolution. In addition, we have updated the data in Figure 1 and Figure S1 to be suitable for the readers.

Minor revisions:

1.P. 1, first paragraph: Delete the phrase “required by them” at the end of the 3rd sentence. The following sentence is vague and possibly unnecessary.

Response 1: Thank the reviewer for pointing out our inaccurate statement. The sentence has changed as follows: 

In particular, amine transaminases (ATAs) have become the most prominent biocatalysts for the generation of optically pure chiral amines because of their high stereoselectivity and environmentally mild reaction conditions 

Reviewer 2 Report

My major concern relates to the very limited timescale of the MD simulations (only 10 ns). The analyses of the trajectories have been carried out in the last few nanoseconds. There are no evidences that a sort of equilibrium has been reached in the simulation trajectories (monitoring for example some global parameters such as RMSD or gyration radius).

The convergence of the simulation has not been assessed either.

Considering current standards, the reported simulation timescales seem are by far too short.

The authors are kindly requested to extend the simulations to at least 100-200 ns and the analyses on the trajectories structures. If not, at least, they should mention this limitation (10 ns) and justify it.

The authors should highlight the agreement between the crystallographic and MD data on the protein dynamics: the authors could derive the RMSF values from the simulation and compare them to the average B-factors, namely of the residues prompted by the Mutual Information Server to Infer Coevolution.

The thermostability of the mutants would need a more thorough biophysical characterization by: i)  circular dichroism spectroscopy  and ii)  differential scanning fluorimetry (Thermofluor)

IMHO the measurement of just the % of the residual activity is not very informative on the likely structural / conformational modifications.

Author Response

Reviewer #2: My major concern relates to the very limited timescale of the MD simulations (only 10 ns). The analyses of the trajectories have been carried out in the last few nanoseconds. There are no evidences that a sort of equilibrium has been reached in the simulation trajectories (monitoring for example some global parameters such as RMSD or gyration radius).

The convergence of the simulation has not been assessed either.

Considering current standards, the reported simulation timescales seem are by far too short.

The authors are kindly requested to extend the simulations to at least 100-200 ns and the analyses on the trajectories structures. If not, at least, they should mention this limitation (10 ns) and justify it.

Response 1: Thank the reviewer for good suggestion. The wild-type and mutant (L118T) AT-ATA were selected for molecular dynamic simulation, which was performed at 313 K and 400 K for 10 ns using the YASARA (version 16.4.6) software with Amber 14 force field [24]. Molecular simulations were also performed for 10 ns in several references published by our research group (Huang et al, 2017; Xie et al., 2018; Huang et al., 2015). Due to the limitation of revision time with my responses to the reviewers as soon as possible, it will take about 50 days to complete the MD simulation for 100 ns.

Huang J, Xie DF, Feng Y. Engineering thermostable (R)-selective amine transaminase from Aspergillus terreus through in silico design employing B-factor and folding free energy calculations. Biochemical and Biophysical Research Communications, 2017, 483(1): 397-402.

Xie DF, Fang F, Mei JQ, et al. Improving thermostability of (R)-selective amine transaminase from Aspergillus terreus through introduction of disulfide bonds. Biotechnology and Applied Biochemistry, 2018, 65(2): 255-262.

Huang J, Jones BJ, Kazlauskas RJ. Stabilization of an α/β-hydrolase by introducing proline residues: salicylic binding protein 2 from tobacco. Biochemistry, 2015, 54(28): 4330-4341.

2. The authors should highlight the agreement between the crystallographic and MD data on the protein dynamics: the authors could derive the RMSF values from the simulation and compare them to the average B-factors, namely of the residues prompted by the Mutual Information Server to Infer Coevolution.

Response 2: Thank the reviewer for good suggestion. Putative important residues in amine transaminase (AT-ATA) identified by conservation, mutual information (MI), cumulative MI (cMI), and proximity MI (pMI) scores from the PF01063 MI network. The MI network represents the interaction between the residues; cumulative Mutual Information (cMI) that measures the degree of shared mutual information of a given residue and the proximity Mutual Information (pMI), which tells about the networks of mutual information in the proximity of a residue (distance < 5 Å) (Figure S1B); The high conservation degree to a residue position was related to slowly changing during protein evolution. The B-factors value of residues were not considered by the Mutual Information Server to Infer Coevolution.

1. The thermostability of the mutants would need a more thorough biophysical characterization by: i)  circular dichroism spectroscopy  and ii)  differential scanning fluorimetry (Thermofluor)

Response 3: Thank the reviewer for good suggestion. The corrections in the revised manuscript are as follows:

4.8.2. Differential scanning fluorimetry

Differential scanning fluorimetry (DSF) is an inexpensive and rapid method to identify the stabilized purified proteins. The wild-type and the best mutant (L118T) AT-ATA were analyzed using the method of Niesen et al. [32], with minor modifications.

The mixture consisted of 100 μg/mL protein samples, SYPRO Orange dye 1x, 150 mM NaCl and 50 mM Sodium phosphate reaction buffer (pH 8.0), with a total volume of 50 μL. A negative control tube containing dH2O instead of protein were included. Subsequently, samples were monitored by StepOne Plus™ Real-Time PCR System (version 2.2.2). Temperature was increased from 25 to 70 °C in increments of 0.7 °C, with each temperature being held for 30 sec. Excitation and emission wavelengths were 490 and 605 nm, respectively.

The melting temperature (Tm) of AT-ATA was calculated using the following equation:

(1)

where UF and NF are the values of minimum and maximum fluorescence intensities, respectively, and a denotes the slope of the curve within Tm.

In DSF, the fluorescence intensity was plotted as a function of temperature (Figure 4). The Tm values for wide type and the L118T muant were 41.37 ± 0.19 °C and 46.37 ± 0.04 °C, respectively.

Round 2

Reviewer 1 Report

Title: A single mutation increases the thermostability and activity of Aspergillus terreus amine transaminase

The revised manuscript is improved by the inclusion of DSF data and correct descriptions of LC-MS experiments. However, the claim that hydrogen bonding to the side chain of L118T is responsible for enhanced activity and stability is still present and even reinforced in the revision. The observation that L118A, L118I and L118V mutants also exhibit enhanced activity and stability directly contradicts this assertion. As such, this conclusion should be removed or the discrepancy at least addressed. I also do not understand the resistance of the authors to acknowledge and reference the existing AT-ATA crystal structure in the main text of the manuscript, especially as this structure first appears in Figure 1. Descriptions of MISTIC appear to be taken verbatim from the reference (Simonetti et al, Nucleic Acids Research 41, W8-14). Specifically:

In the Intro: “MISTIC allows for a comprehensive, compact, visually rich view of the information contained within an MSA”

In the Results: “cumulative Mutual Information (cMI) that measures the degree of shared mutual information of a given residue and the proximity Mutual Information (pMI), which tells about the networks of mutual information in the proximity of a residue”

While I do not feel this is intentional plagiarism, the text should be modified.

Finally, the revision also introduced some minor text errors:

1.     Abstract, change “an increasement of” to “increased”

2.     Figure 4 legend, replace DFS with DSF

3.     Line 201, change “increasement” to “increase”

Reviewer 2 Report

I have very much appreciated the authors' efforts to address one of the issues raised by this Reviewer, i.e the request for a better biophysical characterization by differential scanning fluorimetry of the WT_AT-ATA and of the mutants.

The authors have now at least included (see Figure 4) and compared the DSF profiles of the WT_AT and that of the best performing mutant L118T. The results presented although incomplete (data have not been presented for the other mutants, i.e L118A, L118I and L118V) are nevertheless fairly convincing.

Which are the specific activity (U/mg) of the L118T, L118A, L118I and L118V mutants?

Instead, I am rather baffled by the authors' responses to the concerns of this Reviewer on the very limited timescale of the MD simulations (only 10ns). 

The analyses of the trajectories have been carried out in the last few nanoseconds (it should have been pointed out how many nanoseconds have been considered for the analyses). 

There are no clear evidences whether the system has reached the equilibrium i.e. by monitoring some global parameters such as the RMSD or the radius of gyration.

The convergence of the simulation has not been assessed either.

Considering current computational standards, the reported simulation timescales are by far too short.

The authors are kindly requested to extend the simulations to at least 100-200ns in order to carry out a meaningful analysis of the resulting MD trajectories and to draw solid conclusions.

Indeed the previously three published MD simulations (2ns, 10ns and 10ns respectively) by the authors (see Response 1) IMHO should be carefully revisited.

The agreement of the per-residue average B-factors profile of the crystal structure (PDB ID 4CE5) and the RMSF profile from the MD simulations should be highlighted and thoroughly discussed.

In conclusion, it is with great regret that this Reviewer has to express his concerns in considering the revised manuscript, at least in the present form, to be suitable for publication in Molecules.